# Retinal miRNA Functions in Health and Disease

**DOI:** 10.3390/genes10050377

**Published:** 2019-05-17

**Authors:** Marta Zuzic, Jesus Eduardo Rojo Arias, Stefanie Gabriele Wohl, Volker Busskamp

**Affiliations:** 1Center for Regenerative Therapies Dresden (CRTD), Technische Universität Dresden, 01307 Dresden, Germany; marta.zuzic@tu-dresden.de (M.Z.); Jesus_Eduardo.Rojo_Arias@tu-dresden.de (J.E.R.A.); 2Department of Biological and Vision Sciences, The State University of New York, College of Optometry, New York, NY 10036, USA; swohl@sunyopt.edu

**Keywords:** microRNA, retina, photoreceptors, rods, cones, bipolar cells, Müller glia, retinal inherited disorders, retinitis pigmentosa, retinal degeneration

## Abstract

The health and function of our visual system relies on accurate gene expression. While many genetic mutations are associated with visual impairment and blindness, we are just beginning to understand the complex interplay between gene regulation and retinal pathologies. MicroRNAs (miRNAs), a class of non-coding RNAs, are important regulators of gene expression that exert their function through post-transcriptional silencing of complementary mRNA targets. According to recent transcriptomic analyses, certain miRNA species are expressed in all retinal cell types, while others are cell type-specific. As miRNAs play important roles in homeostasis, cellular function, and survival of differentiated retinal cell types, their dysregulation is associated with retinal degenerative diseases. Thus, advancing our understanding of the genetic networks modulated by miRNAs is central to harnessing their potential as therapeutic agents to overcome visual impairment. In this review, we summarize the role of distinct miRNAs in specific retinal cell types, the current knowledge on their implication in inherited retinal disorders, and their potential as therapeutic agents.

## 1. miRNA Biogenesis and Function

MicroRNAs (miRNAs) are a class of highly conserved ~22 nucleotide (nt)-long non-coding RNAs that have a repressive impact on gene expression in a sequence-specific manner (Figure 1). These miRNAs are derived from partially complementary primary RNA transcripts (pri-miRNA) produced mainly by RNA polymerase II, but also by RNA polymerase III. Pri-miRNAs self-anneal to form hairpin or stem–loop structures, which are subsequently cleaved 11 base pairs (bp) from the base of the hairpin stem by the miRNA-processing complex, a protein ensemble that contains the Drosha ribonuclease and the DiGeorge Critical Region 8 (Dgcr8) protein [1]. The resulting 70 nt-long sequence is known as precursor miRNA (pre-miRNA) and is characterized by a 5’ phosphate and a 2-nt overhang at the 3’ end [2]. Alternatively, miRNAs also result from splicing events independently of Drosha and Dgcr8 [3]. Pre-miRNAs are subsequently exported out of the nucleus for the next processing step in which the Dicer endoribonuclease creates a cut ~22 nt away from the cleaving site of the miRNA-processing complex. Thereby, an additional 5’ phosphate and a new 2-nt 3’ overhang are generated at the opposite end of the double-stranded RNA [4]. The final miRNA structure is thus a duplex formed by partially complementary 22 nt-long RNA sequences with 5’ phosphates and 2-nt 3’ overhangs at both ends. In the last processing step, these duplexes are incorporated into the Argonaute protein, which is a member of the RNA-induced silencing complex (RISC), where one of the strands is removed. The strand that remains bound to Argonaute coordinates the search of, and pairing to, partially complementary target mRNA transcripts [1,5]. Upon binding their mRNA targets, miRNAs either induce their degradation, promote their deadenylation, or reduce their translational efficacy [6]. As individual miRNAs can have thousands of mRNA targets, they represent potent regulators of gene expression at the systems level [7,8]. To date, there are 1917 annotated human miRNA sequences (miRbase v22, access date 2019-05-13) [9].

Dysregulated miRNA expression during development can lead to severe defects and is connected to many pathological conditions such as cancer, neurodegenerative diseases, heart failure, diabetes, and inherited genetic disorders [10]. In mature organs and tissues, miRNAs support the robustness of gene networks and buffer against the fluctuations in gene expression often resulting from stochastic modulation and environmental stress [11]. As their metabolic activity is considerably high and they are often exposed to high levels of external stress, photoreceptors are highly vulnerable to cell death in retinal degenerative diseases. In this context, miRNAs play an important role in photoreceptor survival and function [12]. Moreover, as not all retinal disease phenotypes have been linked to specific genes, altered miRNA expression may underlie the emergence and progression of certain retinal disorders. Thus, exploring the function and gene regulatory networks of these and other non-coding RNAs is of the utmost importance. A dominant mutation in miR-204, for instance, is the genetic cause of retinal degeneration associated with ocular coloboma, a genetic developmental disorder characterized by keyhole-shaped defects in various eye structures [13]. It is thus tempting to speculate that miR-204 is not the sole case of a miRNA giving rise to an ocular disease, and that altered expression of non-coding RNAs might lead to other retinal disorders. Here, we discuss some of the known functions of miRNAs in the adult retina.

## 2. Controlling Cellular miRNA Expression

The deliberate modulation of gene expression through the use of non-coding RNAs, commonly referred to as RNA interference [14], has been extensively utilized in basic and biomedical research. Small hairpin RNAs (shRNAs), for instance, are artificial RNA molecules with a hairpin region processed by the same machinery as miRNAs and effectively acting as miRNA mimics [15,16]. Normally, shRNAs are designed to match the sequence of specific RNA molecules, which they target and downregulate upon delivery. The production of such shRNA molecules is driven by regular promoter elements within a plasmid or a viral vector. Standard gene delivery techniques can be applied to express the shRNA of interest in target cells. This system can also be used to overexpress specific miRNAs [17]. Alternatively, the consequences of miRNA overexpression can be studied by inserting them within artificial intron systems. In general, shRNA vectors are particularly interesting for silencing dominant disease-causing transcripts and their use in retinal cell types was established already over a decade ago [18].

Vice versa, for downregulating or silencing miRNAs, commonly used techniques include “antagomirs” [19] and so-called miRNA “sponges” [20]. Whereas antagomirs are chemically engineered oligonucleotides, miRNA sponges are RNAs with artificial tandem binding site arrays for specific miRNAs. Hence, these sponge sequences compete with physiological targets for miRNA binding, which ultimately leads to reduced silencing of primary mRNA targets. Sponge cassettes can be delivered to retinal cell types by adeno-associated viruses (AAV) [21], which are powerful clinically approved vectors for ocular gene transfer [22,23]. Thus, the therapeutic use of non-coding RNAs is presently growing at an accelerated pace due to increased knowledge of miRNA functions and the wide adoption of technologies that facilitate control of their expression levels in distinct retinal cell types.

## 3. Photoreceptor–miRNAs as Cell Maintenance and Survival Regulators

Rod photoreceptors are sensory neurons essential to night vision. To investigate the impact of miRNA deficiency in these cells, rod photoreceptor-specific Dicer conditional knock-out (cKO) mice have been generated. As mentioned above, Dicer is the endoribonuclease responsible for the second cleavage step in the canonical miRNA biosynthesis pathway. These mice were generated by crossing a *Dicer1^fl/fl^* line with mice expressing Cre recombinase exclusively in mature rods. Loss of *Dicer1* at postnatal day 28 (P28), a time-point at which rods are mature and postmitotic, was reported to lead to outer segment disorganization in eight-week-old mice, followed by robust retinal degeneration and loss of visual function by 14 weeks. Notably, cKO mice did not exhibit significant defects in either phototransduction or the visual cycle before the onset of retinal degeneration, suggesting that the main role of miRNAs in rods is to support photoreceptor survival [24].

Additional studies have aimed at revealing the functions of miRNAs in cone photoreceptors, which are essential for high-acuity and daylight vision. Cone photoreceptor-specific miRNA-deficient mice have been generated by crossing *Dgcr8^fl/fl^* animals with mice expressing Cre recombinase solely in differentiated cones. In these mice, the Dgcr8 protein was only gradually depleted over time as a consequence of its prolonged half-life. Thus, loss of miRNA processing was first detected at P30 and was complete only by P60. The lack of miRNAs in these animals resulted in the progressive loss of cone outer segments, and therefore in low sensitivity to high light levels. However, cones without outer segments did not degenerate in spite of their severely altered gene expression profiles. The latter suggests a crucial role for miRNAs in regulating genetic pathways essential to cone outer segment maintenance and function, but not to cone survival [17]. On the other hand, a recent study reported that the conditional knockout of Dicer in cones results not only in outer segment loss but also in a more severe phenotype with enhanced cone cell death [25]. For proper phenotype interpretation, the targeted miRNA biogenesis proteins, Dgcr8 versus Dicer, are of importance as Dgcr8 knockouts may have residual miRNA expression from splicing products. Although these reports also differed in the cone-specific Cre driver lines used and in the onset of miRNA loss, together they provide strong evidence for the importance of miRNAs on photoreceptor homeostasis, function, and survival.

## 4. The Impact of the miR-183/96/182 Cluster on Photoreceptors

The miRNAs of the miR-183/96/182 cluster play important functional roles in multiple sensory tissues, as evidenced by their expression not only in the retina [26,27], but also in the inner ear [28], the olfactory and gustatory epithelium [27], and in dorsal root ganglia mechanosensory neurons [29]. miR-183, -96, and -182 are expressed as a single polycistronic transcript and exhibit significant sequence similarity in their seed regions. Thereby, they possess shared targets and can partially substitute each other’s function. This overlap in function explains why targeted deletion of only one of these three miRNAs, i.e., miR-182, results in no visible alterations in retinal development [30]. More importantly, although these three miRNAs possess distinct targets, the majority of such targets are involved in identical pathways [31]. In the retina, the miR-183/96/182 cluster is enriched in rod and cone photoreceptors with transcript levels reduced in dark and increased in light conditions (Figure 2). Such dynamic changes in expression levels are the consequences of rapid miRNA decay and of increased transcription, respectively. The latter suggests that miRNA metabolism, in general, is higher in neurons than in other cell types, possibly due to neuronal activity [21].

The function of the miR-183/96/182 cluster was examined in a transgenic mouse line expressing a miRNA sponge for all three cluster miRNAs exclusively in mature rods [32]. Retinae from these mice displayed no detectable morphological or functional changes, likely because the sponge activity was insufficient to capture all mature miRNA molecules of such a highly expressed cluster. Nonetheless, after 30 min exposure to high light intensities (10,000 lux), transgenic mice but not wild type animals showed severe retinal degeneration, indicating that reduced levels of cluster miRNAs have an impact on rod function and survival. More severe retinal defects and retinal degeneration were observed in a miR-183/96/182 cluster knockout mouse model [33], although the miRNAs of the cluster were also missing during photoreceptor development, when cluster expression is tightly controlled [34]. Hence, in this case it is impossible to distinguish the functions of the miR-183/96/182 cluster in development from those in adulthood. Nevertheless, together these knockdown and knockout studies suggest that this cluster plays a neuroprotective role in the retina. Remarkably, re-introducing miR-182 and -183 in Dgcr8 cKO cones, i.e., cones depleted of all other miRNAs, has been reported to prevent the loss of cone outer segments [17]. Considering that miR-182 and -183 constitutes around 70% of all cone miRNA content, it is likely that these play major roles in cone functionality, including in the modulation of outer segment maintenance. Furthermore, overexpression of the miR-183/96/182 cluster in embryonic stem cell-derived retinal organoids induces the formation of light-responsive short outer segments [17], while in human RPE cells in vitro it triggers their reprogramming to neurons [35]. In this sense, a number of the cluster’s targets have been validated, including pro-apoptotic genes, like *Casp2* [32], genes important for survival such as *Rac1* coding for the small GTP-binding protein [36], and neurotransmitter transporters like the voltage-dependent glutamate transporter *Slc1a1* [21] as well as the sodium- and chloride-dependent glycine transporter *Slc6a9* [36]. The latter, together with the neurogenic effects of overexpressing the miRNAs within this cluster in different cell types, suggests that the miR-183/96/182 cluster is central to photoreceptor homeostasis and serves as a pro-survival factor in stress conditions.

## 5. miR-124 Protects Photoreceptors from Apoptosis

MiR-124 is enriched in neurons of the central nervous system [37,38] including the retina [39], and is one of the most well studied miRNA species. In the retina, miR-124 is expressed in all neuronal cell layers, but most prominently in photoreceptor outer (OS) and inner segments (IS). As both humans and mice possess three miR-124 loci, the generation of a complete miR-124 knockout model has been challenging. Recently, a full miR-124 KO (i.e., all six genomic copies) was achieved in a human induced pluripotent stem cell model of neurogenesis [8]. Neurons generated from KO cells displayed altered morphological and functional features and decreased long-term viability, pointing to the importance of miR-124 in neuronal survival. Of note, among the three miR-124 paralogs (a-1, a-2, and a-3), miR-124a-1 has been previously identified by in situ hybridization as the predominant form expressed in the developing mouse retina, with miR-124a-2 being detected at very low levels and miR-124a-3 expression being almost negligible [40]. In agreement with this, knocking out the miR-124a-1 host gene, i.e., the retinal non-coding RNA 3 (*Rncr3*), abolishes miR-124 expression almost entirely in the mouse cone photoreceptor layer. Additional consequences of miR-124a-1 KO included cone mislocalization and apoptosis, decreased expression of cone-specific genes, and reduced light-responsiveness [40].

A validated target of miR-124a in the retina is *Lhx2*. Indeed, the miR-124a-mediated downregulation of *Lhx2* mRNA levels is necessary for cone survival [40]. As depletion of Dgcr8 in adult cones does not cause cone degeneration, it is very likely that miR-124 is effective in these cells early during development but that after differentiation they remain unaffected by miR-124 deletion. In agreement, neurogenesis is hindered neither in the complete miR-124 KO in vitro model nor in miR-124a-1 KO mice, although increased apoptosis was observed over time in both cases. Moreover, as cones but no other retinal neurons degenerate in miR-124a-1 KO mice, it is possible for the compensatory effects of the other two miR-124 paralogs to be insufficient in these photoreceptors but more potent in other retinal cell types.

In the retina, miR-124 is expressed not only in photoreceptors but also in neurons within the inner nuclear layer and the ganglion cell layer. An altered distribution of miR-124 expression from primarily the outer nuclear layer to the inner nuclear layer has been reported to occur in age-related macular degeneration (AMD) patients and mouse models of retinal degeneration [41], with such changes being followed by miR-124 depletion at later degeneration stages. Moreover, environmental stress factors, such as high light intensities, are speculated to induce the translocation of this miRNA from photoreceptors to Müller glia (MG) cells, where it targets the CC-chemokine ligand 2 (*Ccl2*). Ccl2 is a pro-inflammatory chemokine that attracts microglia/macrophages and that is produced and released by MG cells in response to retinal damage [42]. Further, this chemokine is upregulated in both neovascular and atrophic forms of AMD [43] and in retinitis pigmentosa [44]. By downregulating *Ccl2*, miR-124 translocation would thus dampen the inflammatory cascade and promote photoreceptor survival. After prolonged stress, on the other hand, miR-124 depletion results in highly pro-inflammatory environments and thereby in subsequent photoreceptor degeneration. Supporting these observations, *Ccl2* downregulation has been shown to reduce photoreceptor cell death in animal models of retinal degeneration [44,45,46]. Notably, intravitreal administration of miR-124 mimics decreases retinal inflammation and photoreceptor cell death, while preserving retinal function [41]. The latter might be a consequence, at least partially, of increased *Ccl2* targeting.

## 6. miRNA Functions in Inner Retinal Neurons

For the maintenance and survival of retinal cell types other than photoreceptors, miRNAs are also remarkably important. For example, miR-125b has been described as a regulator of neuritogenesis during remodeling in rod bipolar cells after retinal degeneration [47]. Similarly, in retinal ganglion cells (RGCs), miRNA expression profiles and functions have been investigated. RGCs are responsible for sending the visual information collected by photoreceptors to higher brain areas via the optic nerve, and their damage is a hallmark of glaucoma. In retinae of a glaucoma mouse model, nine miRNAs (out of 17 tested) were identified as differentially expressed relative to controls. Among those nine miRNAs, the pro-apoptotic miR-16, -497, -29b, and let-7a were downregulated, while the anti-apoptotic miR-27a was upregulated. Whereas let-7a exerts its apoptotic function by inducing neurodegeneration via Toll-like receptor signaling, miR-16, -29b, and -497 negatively regulate the apoptosis regulator Bcl-2. These alterations in miRNA expression profiles suggest a shift towards a protective anti-apoptotic phenotype [48]. In a different mouse model of glaucoma, downregulation of miR-149 led to an increased RGC number and minimized ultrastructural RGC alterations [49]. In this study, betacellulin (*Btc*) was also identified as a miR-149 target. BTC is a mitogen influencing the activation of the PI3K/AKT signaling pathway, which mediates RGC protection via its pro-survival and anti-apoptotic effects. Further, in the N-methyl-D-aspartate (NMDA)-induced glaucoma mouse model, significant reductions in RGC viability were accompanied by miR-93-5p downregulation [50]. Reduced miR-93-5p levels in these mice, in turn, resulted in the elevated presence of its target phosphatase and tensin homolog (PTEN), which promotes autophagy of RGCs via the AKT/mTOR pathway. Confirming the central role of miR-93-5p in this context, the reduced viability of RGCs isolated from retinae of NMDA-induced glaucoma animals was counteracted either by overexpressing this miRNA species or by transfection with a miR-93-5p mimic.

In mouse models of optic nerve injury, increased expression of specific miRNAs has also been detected. miR-21 expression, for instance, correlated with reactive MG gliosis. Whereas activated MGs are neuroprotective after injury in the acute response phase, they later acquire a pro-inflammatory and pro-apoptotic phenotype. Modulating MG gliosis in both the acute and chronic post-injury phases in these mice resulted in enhanced RGC survival and functionality and led to improvements in retinal structural integrity [51]. Often, however, the protective roles of distinct miRNAs in RGCs have been assessed in vitro. In rat RGC-5 cells, for instance, overexpression of miR-187 was reported to suppress apoptosis and promote proliferation by targeting *Smad7* [52]. Similarly, upregulation of miR-211 was shown to downregulate *Frs2* and to decrease the extent of cell death in RGC-5 cells subjected to a high-pressure challenge [53]. These results hint at a protective effect of miR-187 and miR-211 on RGCs. On the other hand, lentivirus-mediated down-regulation of miR-100 has been described to reduce hydrogen peroxide-induced RGC apoptosis and to enhance neurite growth by activating the AKT/ERK and TRKB pathways through phosphorylation [54]. In a similar experimental approach, hydrogen peroxide-induced apoptosis of RGC was described to decrease upon miR-134 downregulation. In this case, a luciferase reporter assay confirmed that miR-134 directly interacts with the cyclic AMP-response element-binding protein (*Creb*), a transcription factor with central roles in neuronal protection that modulates the expression of the anti-apoptotic proteins BDNF and Bcl-2. Thus, inhibiting miR-134 effectively reduces apoptosis levels by allowing the enhanced translation of CREB and, consequently, the upregulation of its downstream targets BDNF and Bcl-2 [55].

## 7. The role of miRNAs in Müller Glia Development and Function

MiRNAs play an important role in retinal development as described in several studies using Dicer conditional knockouts [56,57,58] (summarized in [59]). One study in particular reported that the blockade of miRNA genesis in an αPax6-Cre mouse model prevented the development of late retinal progenitors and their progenies, including MG [60]. Specifically, three miRNAs have been identified as key regulators of the early to late developmental transition in retinal progenitors, namely let-7, miR-125, and miR-9 [61,62]. Thus, miRNAs may be involved in MG differentiation. As mentioned above, the miRNA expression of whole retinae has been previously profiled [63,64]. Since the vast majority of retinal cells are rod photoreceptors and MG account for only around 2% of them [65], the profiling and identification of MG-specific miRNAs requires FACS-purification or primary cultures. Two independent studies have so far aimed at identifying these miRNAs. The authors of the first study isolated MG from P8 retinae and cultured the cells before miRNA profiling. The culture period, however, was not clarified. In this study, miR-143, miR-145, miR-214, miR-199a, miR-199b*, and miR-29a were identified as highly expressed [66]. In the second study, miR-204, miR-125b, and miR-9 were identified as MG-specific miRNAs (mGliomiRs) in reporter-labelled (RlbpCreERT-dtTomato) FACS-purified P11/P12 murine MG [67]. These three miRNAs had high expression levels in MG, with increasing levels from P11 to adulthood. Moreover, besides these MG-specific miRNAs, a distinct set of miRNAs had similar expression levels in both neurons and glia (termed shared miRNAs). This set includes most members of the let-7 family and miR-29a. Thus, it is conceivable that miR-125, miR-9, and let-7 are not only key regulators in the early to late progenitor transition [61], but that they are also important for MG maturation and function.

A remaining question was whether miRNAs are required for proper MG function. This was addressed in a Dicer cKO study using a MG-specific Rlbp-CreERT:tdTomato:Dicer^fl/fl^ reporter mouse [68]. Dicer deletion was induced in these mice by tamoxifen administration at P12, when MG are differentiated, as well as in one-month old animals. Over a period of 6–12 months, glia aggregated abnormally, leading in turn to massive structural disturbances. Although MG usually do not express the *Bcan* gene, which encodes a chondroitin sulfate proteoglycan, RNA-Seq showed this to be the most highly upregulated gene in these cells. Notably, this phenotype was primarily the consequence of miR-9 loss. miR-9 targets the 3’UTR of *Bcan*, and its administration in vitro or ex vivo prevented abnormal glia development and/or partially restored overall retinal structure. Although in these mice miRNAs were only deleted in MG and not in neurons, a massive loss of photoreceptors was also observed. More importantly, retinal remodeling also subsequently ensued. Remodeling generally occurs as a result of photoreceptor degeneration [69]. Interestingly, MG from human retinitis pigmentosa tissue displayed similar cell aggregation and were positive for Brevican (the protein encoded by *Bcan*). This suggests that not only loss of or disturbances in neuronal miRNAs, but also dysregulation of the glial miRNA biogenesis machinery could be causative of retinal diseases [68]. This hypothesis is further supported by MG ablation studies showing that the dysfunction of this cell population plays an important role in retinal diseases [70,71,72,73].

## 8. miRNAs in Müller Glia De-Differentiation and Their Potential Regeneration Capacity

Although there are miRNAs specific to retinal progenitors, neurons, and MG, and despite many of these miRNAs playing important roles in cell fate decisions, their capacity to alter the fate of mature, fully-differentiated cells remains controversial. In contrast to mammalian retinae, the fish retina can regenerate completely after injury. Remarkably, MG are the cells that mediate the regeneration process. MG de-differentiate to progenitors, proliferate, migrate, and differentiate de novo into all retinal cell types, including RGCs and photoreceptors. Key regulators in this regenerative process are the Acheate-scute family bHLH transcription factor 1 (*Ascl1*) and the RNA-binding protein Lin-28, which are both regulated by the miRNA let-7 [74,75]. Additional factors discovered to be involved in this process include Wnt [76] and Shh [77], which are regulated by the miRNA let-7 in fish and mouse. Notwithstanding, the mechanisms of MG reprogramming in mice, especially at adult ages, are more intricate and involve epigenetic alterations [78].

The brain-enriched miR-124 is known to regulate neurogenesis [39,41,67,79]. Accordingly, overexpression of miR-124 in MG-derived mouse progenitors in vitro results in twice as many βIII-Tubulin-positive cells as in wild-type control cells [66]. This indicates that this neuronal miRNA can be used to direct retinal progenitor like-cells towards a neuronal fate, as it has previously been observed on neural stem cells [80,81]. Moreover, when miR-124 alone or in combination with miR-9/9* was overexpressed in P12 mouse MG in vitro, MG de-differentiated into *Ascl1*-expressing progenitor-like cells that later on expressed mature neuronal markers including *Map2*, Calbindin, and Calretinin. The underlying mechanism for this process is known to require the RE1-silencing transcription factor (Rest) pathway [67], an evolutionarily conserved transcriptional regulator that inhibits neuronal gene expression in non-neuronal cells such as glia or fibroblasts [82,83,84], and whose inhibition by miR-124 enhances the expression of neuronal genes [85,86,87]. In addition, miR-124 has recently been described to promote axon growth in RGCs differentiated from young rat retinal stem cell-derived MG by silencing the REST complex [88]. Besides miR-124, miR-28 has also been reported to potentially play a role in photoreceptor differentiation. MG-derived progenitors treated with anti-miR-28 exhibited *Crx* and Rhodopsin expression while miR-28 overexpression did not [89]. In this case, however, the underlying mechanisms remain elusive.

## 9. Global miRNA Alterations in Retinal Diseases

Retinitis pigmentosa (RP) is a complex inherited retinal disease that emerges from a heterogeneous pool of mutations [90]. Yet, irrespective of the underlying mutation, clinical manifestations are often similar and include progressive photoreceptor loss and visual impairment. To determine if miRNA dysregulation is involved in the pathophysiology of RP, miRNA expression levels have been interrogated in multiple mouse models of retinal degeneration. One study examined the miRNA expression profile of two rhodopsin (*Rho*) mutants (recessive rho–/– knockout and dominant P347S-Rhodopsin, also known as R347) and two RDS/Peripherin mutants (recessive rds–/– null mutant and dominant Δ307-rds) [91]. The authors reported all four mouse models to exhibit miR-96, -182, and -183 downregulation, and miR-1, -133, and -142 upregulation. The dysregulated miRNA signature of isolated rods as well as of whole retinae of R347 and Δ307-rds mice were similar. These findings suggest that altered miRNA profiles are indeed associated with RP, irrespective of the causative mutation or of its dominant or recessive nature. However, the miR-183/96/182 cluster is highly expressed in photoreceptors and therefore its downregulation might be a consequence of the massive rod photoreceptor degeneration rather than causative of RP. In a follow-up study, C-terminal Binding Protein 2 (*Ctbp2*) was validated as a miR-1 target, with *Slc6a9* and *Rac1* being recognized by miR-9, -182, and -183 [36]. CTBP2, a major component of specialized synapses, was shown to be co-expressed with the miR-1/133 cluster in photoreceptors, and its levels were reported to decrease in photoreceptor synaptic regions of R347 retinae. The latter suggests that miR-1/133 may play a role in regulating synaptic remodeling at photoreceptor synapses by targeting *Ctbp2*. In contrast, *Slc6a9*, which encodes one of the two main transporters involved in removing glycine from the synaptic cleft, was co-expressed with miR-1/133 exclusively in cone photoreceptors.

The retinal degeneration 10 (rd10) mouse is a commonly used model of autosomal recessive retinitis pigmentosa and has been utilized to study miRNA expression patterns in early retinal degeneration [92]. In these mice, a spontaneous mutation in the rod phosphodiesterase (*Pde*) gene leads to photoreceptor death from post-natal day 16 on as a result of calcium dysregulation [93]. By using microarrays to profile the expression of over 1900 miRNAs in rd10 retinae, 152 differentially expressed miRNAs have been identified after (P17, P19, and P22), but also shortly before (P15) the initiation of photoreceptor apoptosis [92]. The miRNAs identified as differentially expressed prior to the onset of apoptosis include miR-6240, miR-6937-5p, miR-3473b, and miR-7035-5p, which are likely involved in the disease etiology. On the other hand, miR-155-5p, miR-142-5p, and miR-146a-5p were differentially expressed after the onset of apoptosis. Thus, it remains unclear whether these miRNAs counteract or enhance disease progression. By comparing differentially expressed miRNAs with inversely correlated mRNAs and performing gene ontology and biological pathway analyses, a large number of miRNA targets were revealed to encode factors involved in apoptosis, the inflammatory response, calcium signaling, visual perception, and phototransduction.

The differential expression of miRNAs in the context of retinal degenerative diseases has also been studied in canine models. Notably, the early onset retinal degeneration models Xlpra2 (RPGRORF15-microdeletion), Rcd1 (PDE6B-mutation), and Erd (STK38L-mutation), as well as the slowly progressing Prcd (PRCD-mutation) model, have been reported to exhibit similar miRNA expression profiles as healthy control dogs both prior to and at the peak of photoreceptor cell death [94]. Divergences were reported to arise only during the chronic cell death stage, with anti-apoptotic miRNAs such as miR-9, -19a, -20, -21, -29b, -146a, -155, and -221 being upregulated and pro-apoptotic miR-122 and miR-129 being downregulated. Although the miRNA species differentially expressed in canine and murine models of retinal degeneration are not identical, available reports hint at miRNAs playing a role in counteracting the degenerative processes characteristic of diseased retinae by delaying photoreceptor cell death.

A number of additional studies have interrogated the retinal miRNA expression profile in response to injury. Inducing photoreceptor death by ablating MG, for instance, leads to the differential expression of 16 miRNAs, with miR-142, miR-146, and the miR-1/133 cluster in particular exhibiting increased expression levels [95]. In this study, increased miR-133 expression was mainly detected within the outer nuclear layer, where it targets the anti-apoptotic gene cyclin D2. In fact, the cyclin D2 family has been reported to play a neuroprotective role in retinal degeneration [96]. Unexpectedly, a positive correlation between miR-133 and cyclin D2 was also reported, suggesting that miR-133 up-regulates the expression levels of its own target. Further, in models of light-induced photoreceptor death, 37 miRNAs have been described to increase in expression upon light damage, including seven that regulate the inflammatory response (miR-125, -155, -207, -347, -449, -351, -542). Among the latter, miR-155 is of particular interest as it is linked to the progression of the inflammatory response through the targeting of complement factor H (*Cfh*) [97], a major inhibitor of the alternative complement pathway [98]. Moreover, upregulation of miR-155 has been detected in both mouse and canine models of inherited retinal diseases [92,94], suggesting that the roles of certain miRNAs are conserved across species and, more notably, shared in different injury response contexts.

The miRNA profile of human retinae has been recently investigated via high-throughput sequencing [99]. The assessment of miRNA expression levels in 16 retinae from healthy donors led to the detection of 480 miRNA species with more than 3000 variants. Of note, the five most highly expressed miRNAs accounted for 70% and the top 20 miRNAs for almost 90% of the entire retinal miRNA profile. Moreover, the miRNAs miR-182 and miR-183 were the most prevalent miRNA species in retinal neurons and miR-204 was the most abundant miRNA in retinal pigment epithelium (RPE) cells. Some of these highly expressed human miRNAs coincide with those previously identified in murine retinae by microarray analyses and RNA in situ hybridization experiments [27,64,100,101], pointing to their potentially conserved retinal functions across mammalian species [102]. It is known from animal models that miRNA expression patterns change over the course of retinal development [64,101] and that such dynamic miRNA expression profiles are crucial to retinal cell differentiation (reviewed in [59]). Altogether, it is very likely that the impact miRNAs have in animal models is transferable to human retinal cell types in health and disease. This further implies that particular miRNAs may be used as disease biomarkers and, additionally, that the precise modulation of their expression levels might represent a valuable therapeutic strategy for the treatment of diverse retinal inherited disorders.

## 10. Conclusions

We are in the process of understanding the complex functions of miRNAs in retinal health and disease. Further studies are required to obtain coherent pictures of gene expression regulators in order to harness their potential use not only as biomarkers but also to treat and counteract retinal diseases. As individual miRNA species have the potential to orchestrate entire genetic programs, the possibility to use them as master regulators to stabilize or reset the cellular state of neurons and glial cells in the retina is highly promising.

## Figures and Tables

**Figure 1 genes-10-00377-f001:**
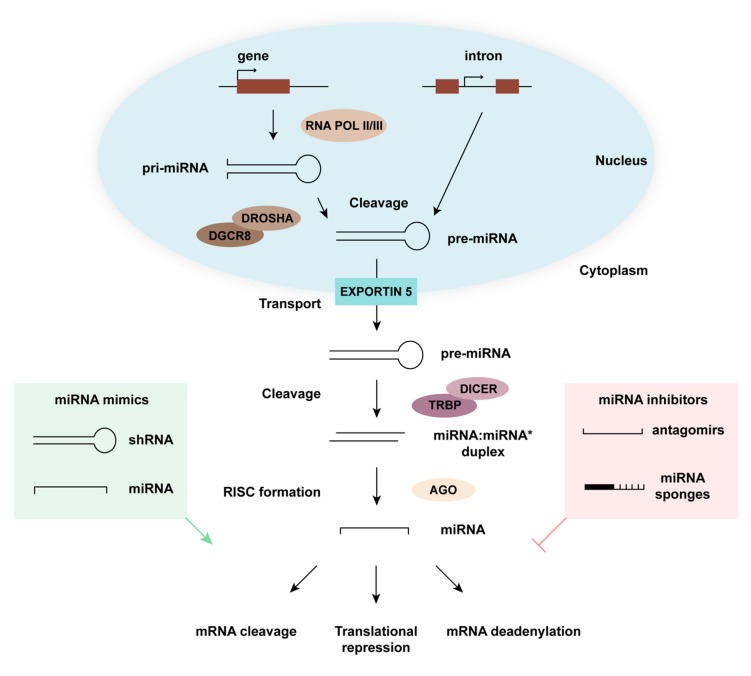
MicroRNA (miRNA) pathway. miRNA biogenesis starts in the nucleus where partially self-complementary RNA polymerase II or III transcripts from a miRNA gene or from an intronic region assemble into a hairpin-like structure known as primary (pri)-miRNA. Pri-miRNAs are cleaved by the DiGeorge Critical Region 8 (Dgcr8)/Drosha complex and transported to the cytoplasm via Exportin 5. In the cytoplasm, pre-miRNAs are cleaved by the Dicer/HIV-1 TAR RNA binding protein (TRBP) nuclease complex, thereby giving rise to a miRNA duplex. This duplex is then loaded onto the Argonaute (AGO) protein, a component of the RNA-induced silencing complex (RISC), where one of the two strands is discarded while the other serves to search complementary transcripts. Targets bound by miRNAs exhibit reduced translational efficiency, mainly as a consequence of mRNA cleavage or deadenylation. The miRNA pathway can be modulated by introducing miRNA mimics (green) or miRNA inhibitors (red). shRNA is small hairpin RNA.

**Figure 2 genes-10-00377-f002:**
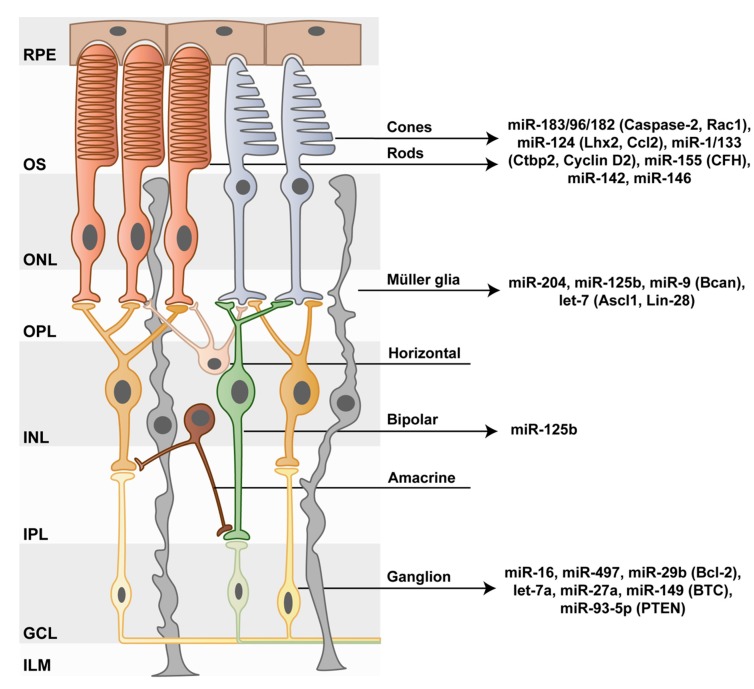
MiRNAs acting as modulators of retinal cell behavior. In the vertebrate eye, the retinal pigment epithelium (RPE) separates the retina from the subretinal space. Within the outermost layer of the retina, rod and cone photoreceptors sense light with their outer segments (OS). Photoreceptor bodies reside within the outer nuclear layer (ONL), and their axons protrude into the outer plexiform layer (OPL), where they synapse with excitatory bipolar cells and inhibitory horizontal cells. The bodies of these cells, as well as of amacrine cells, which create inhibitory synapses with the axons of bipolar cells, reside in turn within the inner nuclear layer (INL). The electrochemical signal produced by photoreceptors during phototransduction is transmitted through bipolar cells to ganglion cells via synaptic connections in the inner plexiform layer (IPL). In a final step, ganglion cells send this information to higher brain areas through their axons, which bundle up to form the optic nerve. An additional cell type within the retina is Müller glia. These cells play key roles in the support of neuronal functions and in mediating the reaction to a number of physiological signals, including immune responses. Müller glia feet form the outer limiting membrane (OLM), which separates photoreceptor OS from their somata. Within the retina, miRNAs play central roles in health and disease. Recognized miRNA species associated to the functionality of specific retinal cell types are shown with validated targets between parentheses. GCL, ganglion cell layer; ILM, inner limiting membrane.

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
