# Peer review of "Retinal miRNA Functions in Health and Disease"

_genes, 2019, doi:10.3390/genes10050377_

Round 1
Reviewer 1 Report
Dear authors,
congratulations to this well written manuscript that nicely summarizes our knowledge about miRNAs in the retina.
I have nothing to comment and recommend the manuscript for publication.
Author Response
We would like to thank Reviewer #1 for the evaluation of our work and the nice comments.
Reviewer 2 Report
In the present review article, Zuzic and collaborators review the current knowledge on the role of microRNAs (miRNAs) in a number of retinal degenerations, and in different retinal cell types. The review is thorough and really up-to-date but I would like to suggest some modifications that in my opinion would greatly improve the manuscript:
1- The current organization of the review is quite confusing, at times redundant and, unfortunately, this makes the paper very hard to follow. For example, after a brief introduction of miRNA biogenesis the article goes ahead to talk about retinitis pigmentosa, a disease that affects rod photoreceptors (page 3, line 76) but rods are introduced later (page 4, line 161). Similarly, mir-182/96/183 downregulation in photoreceptor degenerations is explained in line 84 but this cluster of miRNAs is introduced later in line 187. Line 201 talks about sponges without introducing them and then, sponges are introduced in line 392, etc. This going back and forth makes the article unnecessarily convoluted and gives the impression that several parts were pasted together instead of reading as a cohesive review. To me, it would make more sense to review first the role of miRNAs in homeostasis of the different cell types to then, transition to alterations in disease models. In any case, I strongly suggest some re-organization to make the review more cohesive and easy to follow.
- Redundancies should be eliminated. For example, lines 88-101 talk about the relationship between mir-182/96/183 but this information is repeated in lines 219-224, etc.
2- I suggest adding some of the most important miRNA targets to figure 2 to help the reader digest the information.
3- The introduction on miRNA biogenesis states that pri-miRNAs are cleaved by the microprocessor (Drosha) but only a fraction of miRNAs are processed through this mechanisms and many pri-miRNAs are actually processed by the splicing machinery. In my opinion this a really important as explains differences between DGCR8 and Dicer mutant models.
4- In my opinion some the interpretation of the data on miR-183/96/182 downregulation in models of degeneration is a bit misleading. As miR-183/96/182 are mostly expressed in photoreceptors and rods represent about 75% of all cells in a murine retina, any rod degeneration will lead to a global loss of miR-183/96/182 but this might just be reflecting the changing cellular composition of the retina and not a direct association between this cluster of miRNAs and retinitis pigmentosa. The sponge data as well as the organoid data indicate that these miRNAs are associated with photoreceptor maturation and homeostasis but I am not sure whether a clear correlation with retinitis pigmentosa can be drawn.
Author Response
In the present review article, Zuzic and collaborators review the current knowledge on the role of microRNAs (miRNAs) in a number of retinal degenerations, and in different retinal cell types. The review is thorough and really up-to-date but I would like to suggest some modifications that in my opinion would greatly improve the manuscript:
We would like to thank Reviewer #2 for the evaluation of our work and the very constructive comments and suggestions.
1- The current organization of the review is quite confusing, at times redundant and, unfortunately, this makes the paper very hard to follow. For example, after a brief introduction of miRNA biogenesis the article goes ahead to talk about retinitis pigmentosa, a disease that affects rod photoreceptors (page 3, line 76) but rods are introduced later (page 4, line 161). Similarly, mir-182/96/183 downregulation in photoreceptor degenerations is explained in line 84 but this cluster of miRNAs is introduced later in line 187. Line 201 talks about sponges without introducing them and then, sponges are introduced in line 392, etc. This going back and forth makes the article unnecessarily convoluted and gives the impression that several parts were pasted together instead of reading as a cohesive review. To me, it would make more sense to review first the role of miRNAs in homeostasis of the different cell types to then, transition to alterations in disease models. In any case, I strongly suggest some re-organization to make the review more cohesive and easy to follow.
- Redundancies should be eliminated. For example, lines 88-101 talk about the relationship between mir-182/96/183 but this information is repeated in lines 219-224, etc.
Response 1: We would like to thank Reviewer #2 for pointing out our imperfections in the previous manuscript organization. We agree with all suggestions and re-organized our revised manuscript accordingly. We now first introduce miRNA sponges “2. Controlling cellular miRNA expression” and moved the disease paragraph to the end “9. Global miRNA Alterations in Retinal Diseases”. We agree with Reviewer #2 that this significantly improved the readability and redundancies were eliminated. Please find all changes tracked in the word file.
2- I suggest adding some of the most important miRNA targets to figure 2 to help the reader digest the information.
Response 2: This is a good point and we have added the most important miRNA targets to Figure 2.
3- The introduction on miRNA biogenesis states that pri-miRNAs are cleaved by the microprocessor (Drosha) but only a fraction of miRNAs are processed through this mechanisms and many pri-miRNAs are actually processed by the splicing machinery. In my opinion this a really important as explains differences between DGCR8 and Dicer mutant models.
Response 3: We fully agree with Reviewer #2 and have updated the text (lines 35/36 and lines 206-208) and updated Figure 1.
4- In my opinion some the interpretation of the data on miR-183/96/182 downregulation in models of degeneration is a bit misleading. As miR-183/96/182 are mostly expressed in photoreceptors and rods represent about 75% of all cells in a murine retina, any rod degeneration will lead to a global loss of miR-183/96/182 but this might just be reflecting the changing cellular composition of the retina and not a direct association between this cluster of miRNAs and retinitis pigmentosa. The sponge data as well as the organoid data indicate that these miRNAs are associated with photoreceptor maturation and homeostasis but I am not sure whether a clear correlation with retinitis pigmentosa can be drawn.
Response 4: Regarding this point, we also agree with Reviewer #2 and have updated the text (lines 418-420).